# Detection of Mycobacterial DNA in Human Bone Marrow

**DOI:** 10.3390/microorganisms11071788

**Published:** 2023-07-11

**Authors:** Alba González-Escalada, María José Rebollo, Jorge Barrios Payan, Rogelio Hernández-Pando, María Jesús García

**Affiliations:** 1Facultad de Ciencias de la Salud, Area of Medical Microbiology, Rey Juan Carlos University, 28922 Alcorcon, Spain; alba.gonzalezescalada@urjc.es; 2Department of Preventive Medicine and Public Health and Microbiology, School of Medicine, Autonoma University of Madrid, 28029 Madrid, Spain; mariajose.l.rebollo@gmail.com; 3Experimental Pathology Section, Department of Pathology, National Institute of Medical Sciences and Nutrition Salvador Zubirán, México City 14080, Mexico; qcjbp77@yahoo.com.mx (J.B.P.); rhdezpando@hotmail.com (R.H.-P.)

**Keywords:** mycobacterial DNA, tuberculosis infection, human bone marrow, adipocytes

## Abstract

Bone marrow is a cell-rich tissue of the reticuloendothelial system essential in the homeostasis and accurate functioning of hematopoiesis and of the immune system; moreover, it is also rich in lipids because it contains marrow adipocytes. This work aimed to evaluate the detection of mycobacterial DNA in human bone marrow as a tool to understand the complex pathology caused by the main pathogen *Mycobacterium tuberculosis* (*Mtb*). Formalin-fixed paraffin-embedded human bone marrow samples were studied using both conventional PCR + hybridization and *in situ* PCR to figure out the cell distribution of the targeted DNA. Samples were retrospectively collected from HIV+ patients with microbiologically proved mycobacterial infection and from subjects without evidence of infection. *Mycobacterium avium* (*Mav*) as well as *Mtb* DNA was detected in both settings, including tissues with and without granulomas. We detected DNA from both mycobacterial species, using *in situ* PCR, inside bone marrow macrophages. Other cell types, including adipocytes, showed positive signals only for *Mtb* DNA. This result suggested, for the first time, that marrow adipocytes could constitute an ideal reservoir for the persistence of *Mtb*, allowing the bacilli to establish long-lasting latent infection within a suitable lipid environment. This fact might differentiate pathogenic behavior of non-specialized pathogens such as *M*av from that of specialized pathogens such as *Mtb.*

## 1. Introduction

Infection with *Mycobacterium tuberculosis* (*Mtb*) rarely leads to disease, the asymptomatic infection being the most frequent early result [1]. According to clinical and epidemiological data, it is considered that this infection without symptoms is not a single stage but a continuum of stages ranging from latent to incipient, then to subclinical, and ending in tuberculosis (TB) disease [2]. During the latent stage, the bacteria are in a dormant condition, mainly inside granulomas [3]. Nowadays, in accordance with the main role of lipids in the crosstalk host–pathogens [4], the relevance of the lipid environment in the establishment and maintenance of that dormant stage by the tubercle bacilli is clear [5]. In humans, lipids mostly concentrate in the adipose tissue distributed in several regions, as, for example, surrounding inner organs or in the skin [6]. Moreover, bone marrow (BM) also holds its own adipose tissue, with characteristic marrow adipocytes, different from the white, brown, or beige adipocytes found in other body locations. In fact, the marrow adipose tissue has a functional specific activity, including secretion of adipokines [7].

BM is a complex cell-rich tissue of the reticuloendothelial system essential in the homeostasis and accurate functioning of hematopoiesis and of the immune system. *Mtb* is an intracellular pathogen with a predilection for invading cells of the reticuloendothelial system and, consequently, for invading organs rich in these cell types, such as BM. In recent years, several studies have been published focused on the study of BM and its role in the persistence and survival of different infectious agents including *Mtb* [8,9,10,11,12,13]. Therefore, BM might be the reservoir in which some microorganisms could set up undetected or chronic infections [5,6,9,14]. For this reason, BM is not only considered a very useful sample for the diagnosis of disseminated infections from *Mtb* or other mycobacteria [15,16], but also its study seems essential to understand mycobacterial pathogenicity, as well as the circumstances that allow those bacilli to establish a successful silent infection [10,11,12,13,17,18].

Despite the known association to lipid environment, marrow adipocytes have not been directly related to *Mtb* infection yet. On the contrary, adipose tissue has been considered an extrapulmonary location for *Mtb* and a potential niche for persistent infection of the bacilli [14,19]. Thus, the high content of triacylglycerol and other lipids would facilitate the survival of *Mtb* inside any type of adipocytes as well as the establishment of latent infection in all those locations [14,20] regardless of the presence of granulomas as described [13].

In the case of *Mtb* infection, it seems that resident stem cells in the BM could constitute an important reservoir for the bacterium, due to the hypoxic, immunoprivileged and antibiotic impermeable environment that exists within them [10,12,17,18]. From this location, and thanks to the close contact between these cells and the extensive network of arterial capillaries in the BM, *Mycobacterium* dissemination may occur to other extramedullary sites where reactivation of the disease may be built up [13,17,18,21]. Dissemination of infections is a relevant issue for immunocompromised patients, as it could happen in people infected with HIV [22]. Co-infection with HIV–mycobacteria remains a main clinical problem, and in many cases, other mycobacteria, such as *Mycobacterium avium* (*Mav*) could employ strategies similar to *Mtb*, as adaptation to hypoxia, to remain inside the host for extended periods of time [23]. *Mav* is one of the most common non-tuberculous mycobacteria found in clinical samples, mainly causing disseminated or extrapulmonary infection in HIV+ people [24]. Similar to *Mtb*, *Mav* also invade and replicate in non-professional phagocyte cells, such as fibroblast or endothelial cells [24]. Furthermore, the pathophysiology of *Mav*-disseminated disease includes its transport through the endothelial reticulum system (particularly the BM), and hence, this organ may play a key role in the development of disseminated infections caused by this bacterium.

In this work, we aimed to perform a retrospective analysis of bone marrow biopsies (BMBs) from subjects putatively infected with mycobacteria, most of them HIV+. We tested those tissue samples for *Mtb* and *Mav* infection to determine the presence of DNA of these two mycobacteria and to figure out their intracellular location by using *in situ* PCR. We detected DNA from the two species, both in people with and without suspicion of mycobacterial infection; moreover, mycobacterial DNA was detected inside different BM cells. Interestingly, only *Mtb* DNA was detected inside marrow adipocytes, stressing the role of lipid environments in the infection of this main pathogen compared to opportunistic pathogens.

## 2. Materials and Methods

### 2.1. Patients and Samples

A total of 54 formalin-fixed paraffin-embedded (FFPE) BMBs from patients treated at Hospital Universitario Doce de Octubre (Madrid, Spain) were analyzed. All BMBs were taken from the posterosuperior iliac spine using clinical standard procedures.

Samples were separated in two groups: group A included 28 biopsies (26 of them HIV positive, 18 with CD4 lymphocyte count <50/mm^3^) with clinical and either positive culture or compatible histology for mycobacterial infection (Table 1); group B comprised 26 control biopsies from HIV negative patients, without any suspicions of mycobacterial infection, whose samples were obtained during monitoring of hematological diseases or solid tumors (Table 2).

### 2.2. Histological and Microbiological Analysis

All BMBs included in the study were fixed in 4% buffered formaldehyde for 24 h, dehydrated in alcohol, cleared in xylene and embedded in paraffin. Three to ten four-micron thick sections were stained with hematoxylin and eosin for histological analysis. As mentioned, biopsies were distributed in group A (from patients suspected of mycobacterial infection) and group B (from patients without suspicion of mycobacterial infection).

Biopsies from group A were cultured using microbiological methods before being embedded. In brief, 0.5–1 mL of fresh BM was incubated in liquid culture media (*Mycobacteria Blood* bottle, BacT-Alert, Bio-Merieux, Madrid, Spain); positive cultures were confirmed using rhodamine acid-fast stain, and mycobacteria were identified using standard tests including specific molecular probes for both *Mtb* and *Mav* (AccuProbe, GenProbe, San Diego, CA, USA). Samples from group B were not included in the microbiological routine analysis because they came from onco-hematological patients without suspicion of mycobacterial infection; therefore, they were embedded directly after collection.

### 2.3. DNA Isolation, Conventional PCR, and Hybridization

DNA was isolated from 10 mm fragments of FFPE biopsies by using the non-ionic detergent procedure [25]. Strict procedures and controls were followed to avoid cross-contamination between samples during DNA extraction. Presence of amplifiable DNA or inhibitors was depicted by amplification of the eukaryotic gene β-globin [26]; only those samples with a positive result for this gene were further considered. Primers used for PCR + hybridization are summarized next: eukaryotic gene β-globin [26] 5′-GAAGAGCCAAGGACAGGTAC-3′ (forward) and 5′-CAACTTCATCCACGTTCACC-3′ (reverse); *Mtb* insertion sequence IS*6110* [27] 5′-CTCGTCCAGCGCCGCTTCGG-3′ (forward) and 5′-CCTGCGAGCGTAGGCGTCGG-3′ (reverse); *Mav* insertion sequence IS*1311* [28] 5′-GGTGCAGCTGGTGATCTCTGA-3′ (forward) and 5′-GTCGGGTTGGGCGAAGAT-3′ (reverse). PCR conditions were applied, as described in the references indicated.

Amplification reactions included the use of dUTP and uracyl-DNA-glycosylase to avoid the contamination carry-over. Contamination at the DNA level was ruled out by using standard procedures of manipulations and by performing PCR analysis in control samples without DNA as a template. All samples were analyzed by standard PCR followed by Southern blot transfer of agarose gels and hybridization (PCR + H) [29]. Amplified products were agarose electrophoresed and Southern blot transferred to nylon membranes (Bio-Rad, Alcobendas, Spain). Membranes were further hybridized by using the following internal oligonucleotides radioactively labeled as probes: IS*6110*, 5′-CACCTATGTGTCGACCTGGGCAGGGTTCGCC-3′; IS*1311*, 5′-GCCGGGTGCACTTCCTGCGCAACGTGCTCG-3′. Stringent conditions were applied for hybridization and washes [29]. Labeled bands were detected using autoradiography (see Results, Figure 1). Amplification and hybridization were performed at least three times per sample. Samples were considered positive when results were positive at least twice.

### 2.4. In Situ PCR

Out of the 54 biopsies in the study, 43 of them were analyzed using *in situ* PCR to detect the cellular location of IS*6110* (20 and 23, respectively, from groups A and B) and 23 were studied for IS*1311* (16 and 17, respectively, from groups A and B) (Table 1 and Table 2). Remaining BMB samples could not be analyzed using *in situ* PCR due to the inability to obtain sufficient histological material.

The *in situ* PCR conditions applied were performed as described previously [30] for detection of IS*6110* and IS*1311* insertion sequences [27,28]. A negative control consisted of performing the whole procedure with non-infected mouse BMBs. Further parallel controls for each test reaction included Taq-polymerase negative and primer negative reactions.

### 2.5. Ethics Statement

This study was approved by the Ethical Committee for Clinical Research of the Hospital Universitario 12 de Octubre (Madrid, Spain).

## 3. Results

*Mtb* DNA was detected by PCR + H in 24 of the 28 group A biopsies. Nine of them corresponded to BM samples in which *Mtb* was isolated by culture, thirteen were biopsies positive for *Mav* in culture (five of them with known previous history of tuberculosis), and two biopsies had negative microbiological culture but granulomas compatible with mycobacterial infection on histological analysis. These two patients had a previous history of tuberculosis or recent contact with a tuberculous patient. Regarding detection of *Mav* DNA, PCR + H was positive in 14 bone marrow samples of group A; in 11 of them, this bacterium had also been isolated using microbiological culture (Table 1).

Some discrepant results were observed when comparing culture and PCR results in group A (Table 1). These results could indicate multiple infections, variable distribution of bacilli in the tissue, or the presence of latent bacilli in the BM of these patients. In some cases, detection of non-culturable bacilli cannot be ruled out (such as BM18 or BM20).

No mycobacterial DNA was detected in three of the samples in group A (BM8, BM24, and BM37) although the histological study suggested tuberculous infection (presence of granulomas), or they belonged to patients with a known previous history of tuberculosis. The culture was negative in two of them (Table 1).

Patients included in group B had no sign or symptom of mycobacterial infection at the time of sample collection. Positive detection of mycobacterial DNA, either *Mtb* or *Mav* (8/26 for each), was obtained in these samples using PCR + H. A review of the clinical records of these patients revealed that five of them have had previous history of tuberculosis, although PCR was positive for *Mtb* in only two (Table 2).

Autoradiography of Southern blots after PCR and hybridization was performed by using insertion sequences IS6110 of *Mtb* (a) and IS1311 of *Mav* (b) as probes (see Methods for more explanation). (a) Line C+ corresponds to positive control (*Mtb* strain 79500 DNA), line C- corresponds to negative control (distilled sterile water), lines 2 (BM2), 4 (BM4), 5 (BM5), 7 (BM7), 9 (BM9), and 11 (BM12) showed positive results; lines 3, 6, 8, 10 (BM8, BM24, BM27, BM37), and 12 showed negative results. (b) Line (C+) corresponds to positive control (*Mav* ATCC 25291^T^ DNA), line (C-) corresponds to negative control (distilled sterile water), lines 2 to 6 (BM26, BM27, BM28, BM29, and BM31), 8 (BM4), and 11 (BM9) showed positive results; lines 1, 7, 9 (BM 24, BM 33 and BM 5), and 10 (BM7) showed negative results.

Regarding the results obtained using *in situ* PCR, the *Mtb* genome was detected in 7 of the 20 biopsies of group A (PCR + H positive in all of them) and in 11 of the 23 from group B (8 of them were PCR + H positive) (Table 2). None of group A (0/16) and 7/17 of group B were positive for *Mav* DNA using this procedure (Table 1 and Table 2). The discrepancies detected when comparing conventional versus *in situ* PCR using the same target are a frequent finding in the literature and are usually explained by the differential distribution of the bacilli inside the tissues [31]. The involvement of BM in the course of asymptomatic mycobacterial infections could also explain these results at some stages.

A positive label denoted by *in situ* PCR was characterized by the presence of blue dots in the cellular cytoplasm. *Mtb* was detected mainly inside macrophages and macrophage precursors, although not as part of a granuloma. Positive DNA signals from this bacterium were also visualized inside non-professional phagocytes, such as fibroblasts and endothelial cells (Figure 2a). Furthermore, we detected *Mtb* DNA positive signals inside adipocytes (Figure 2b) in 3/7 and 4/11 positive samples, respectively, from groups A and B (Table 1 and Table 2). By contrast, the positive signals of *Mav* DNA were located only inside macrophages and its precursors (Figure 2c). Interestingly, in all cases, *in situ* PCR positive cells were found in areas of normal histology, without visible granulomas or necrotic areas by histological analysis.

## 4. Discussion

Tuberculosis is, globally, the leading cause of mortality due to a single bacterial infectious agent in humans [32]. In addition, it is estimated that one-fourth of the world’s population is latently infected, with a high percentage of them at risk of developing active tuberculosis (and transmitting it) at some point in their lifetime [33]. To prevent this, it is necessary to understand all the events that occur during tuberculosis infection, which would allow us to develop more effective diagnostic and therapeutic tools in the fight against this disease.

Although many aspects of tuberculosis infection remain to be elucidated, it seems clear that both BM and adipose tissue play an important role in the development of this infection. More studies are needed to understand the involvement of these tissues in this process, which would allow to develop new diagnostic and therapeutic strategies to control this disease. A recent study described the presence of *Mtb* in peripheral blood mononuclear cells in different patients, including asymptomatic HIV-infected patients. In this study, the authors found that the profile of infected cells was different in HIV-infected subjects compared to non-HIV-infected ones [34]. From these results, it seems relevant to study the role of BM tissue both in immunocompetent and immunocompromised patients to improve our understanding of the several phases of the spectrum of the *Mtb* infection.

In this work, we have analyzed FFPE-BMBs from subjects with and without active *Mtb* infection by using a technique of high sensitivity and specificity, such as conventional PCR + H with specific internal probes [35]. Moreover, we have determined the intracellular location of signaled DNA within BMBs. We detected the *Mtb* genome in more than half of the BMBs (32/54; Table 1 and Table 2). Eleven of these biopsies belonged to patients without suspected or active tuberculosis.

Although some of the patients in the study could not be followed up, because they died shortly after bone marrow sampling due to an HIV-related advanced state of immunosuppression, a later review of the medical records of the rest of the participants showed that they had not developed clinical tuberculosis during their lifetime, meaning that they putatively had latent infection in their BM at the time that the sample was taken. These results are consistent with findings published by other authors and reinforce the idea that BM might play a crucial role in the pathogenesis of tuberculosis [10,11,12,13,17,18].

Studies published so far also suggest that BM stem cells are the immunoprivileged cellular niche in which the mycobacterium would remain undetected in a dormant state and from which, through the bloodstream, it could reach other tissues after reactivation under certain circumstances such as immunosuppression [10,11,12,13,17,18,34,36]. Some studies, devoted to diagnostic purposes in tuberculosis, included BM as a sample to test PCR. From those, *Mtb* DNA was detected in a higher proportion in culture/smear negative BM samples compared to other extrapulmonary samples [37,38,39]. This result also suggests that the tubercle bacilli could have BM as a relevant place during their infective process.

A significant proportion of samples in our series belonged to patients with HIV infection (26 out of 54; Table 1). Many of those samples were biopsies positive for *Mav* in culture; for this reason, we also searched for the presence of DNA from the opportunistic mycobacterial pathogen *Mav*. We detected *Mav* DNA in BMBs from both immunocompromised patients as well as from immunocompetent patients with no apparent infection. *Mav* is a microorganism with an environmental reservoir, such as tap water, that implies a constant human contact. It is not surprising that this mycobacterium was the most frequent non-tuberculous mycobacterium causing extrapulmonary infections in patients with or without immunosuppression [40]. Detection of mycobacterial DNA in samples included within group B could be explained by the fact that pathogens with known hematological spread capability could reach bone marrow tissue and then be detected by chance throughout their processing and eventual clearance inside the host. To rule out this possibility, we checked for the detection of DNA from two other common human pathogens, such as *Klebsiella pneumoniae* and *Salmonella enterica*, by using a similar methodology (detection of specific insertion sequences using PCR + hybridization). We detected DNA only in one sample for each pathogen within group B (see Appendix A). Therefore, the detection of mycobacterial DNA could putatively comprise some mycobacterium-specific pathophysiological pathway in the bone marrow tissue. As mentioned, previous results suggested that bone marrow could also play some role in the pathogenesis of *Mav* infection [24]. Further studies are deserved on BMBs from patients infected with this bacterium to understand their physiopathology more comprehensively.

To determine the intracellular localization of *Mtb* inside BM, we performed *in situ* PCR on 43 of the 54 BMBs in the study, showing a positive result in 18 of them (Table 1 and Table 2). In these samples, *Mtb* DNA was detected in areas of apparently normal histology without granulomas, not only inside macrophages, fibroblasts, and endothelial cells but also inside adipocytes. Detection of *Mtb* DNA in non-professional phagocytes and reticuloendothelial cells is in accordance with other results previously reported [12,13,18], showing that cells of different lineages might be simultaneously infected by *Mtb* also in the BM. To the best of our knowledge, this is the first reported time that *Mtb* DNA was detected inside marrow adipocytes. Furthermore, *Mav* DNA was detected only inside macrophages and macrophage precursors, suggesting that detection in other BM cells might mean some more specific intracellular behavior in the *Mtb* pathogenicity. Because our study was carried out on paraffin-embedded biopsies in which there are no viable microorganisms, it is impossible for us to confirm or reject the hypothesis of *Mtb* migration between cells of different lineages.

Studies published so far have demonstrated the presence of putatively latent *Mtb* in peripheral adipose tissue [5,14], but there were no previous reports on the detection of *Mtb* inside marrow adipocytes. Given that this organ contains a high proportion of adipose tissue, it would not be surprising that this location could represent a suitable place to establish a dormant phenotype by the tubercle bacilli. Actually, adipocytes could constitute an ideal reservoir for the persistence of mycobacteria as they constitute specialized cells where *Mtb* could find a lipid environment suitable to establish long-lasting latent infection [14,20,41,42]. It has also been postulated that, inside these cells, *Mtb* could modulate the expression of different inflammatory mediators with a direct effect on the physiology of adipose tissue and on its energy balance [6,7]. All these facts might differentiate the pathogenic behavior of non-specialized pathogens, such as *Mav*, from that of specialized pathogens, such as *Mtb.* As previously mentioned, by means of working with paraffin-embedded BMB, we have not been able to study the effects of the bacteria on the function and homeostasis of the different types of infected cells, the relationships between them, or the aftermath of these circumstances on the persistence of *Mtb* in the bone marrow.

In agreement with previous data, our results are adding the marrow adipocytes to other previously described stakeholders which play some role during tuberculosis, thus reinforcing the complex pathology of this deadly disease.

## Figures and Tables

**Figure 1 microorganisms-11-01788-f001:**
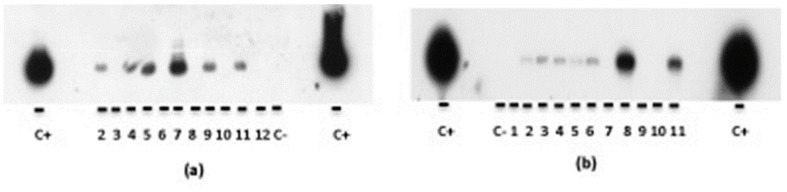
Detection of mycobacterial DNA in bone marrow samples using PCR + hybridization. Autoradiography of southern blots after PCR and hybridization, by using insertion sequences IS6110 of *Mtb* (**a**) and IS1311 of *Mav* (**b**) as probes (see Methods for more explanation). (**a**) Line C+ corresponds to positive control (*Mtb* strain 79500 DNA), line C- corresponds to negative control (distilled sterile water), lines 2 (BM2), 4 (BM4), 5 (BM5), 7 (BM7), 9 (BM9) and 11 (BM12) showed positive results; lines 3, 6, 8, 10 (BM8, BM24, BM27, BM37) and 12 showed negative results. (**b**) Line (C+) corresponds to positive control (*Mav* ATCC 25291^T^ DNA), line (C-) corresponds to negative control (distilled sterile water), lines 2 to 6 (BM26, BM27, BM28, BM29 and BM31) 8 (BM4) and 11 (BM9) showed positive results; lines 1, 7, 9 (BM 24, BM 33 and BM 5) and 10 (BM7) showed negative results.

**Figure 2 microorganisms-11-01788-f002:**
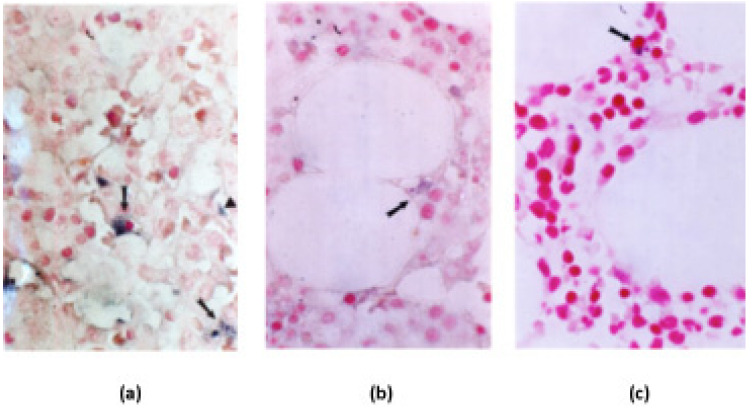
Detection of mycobacterial DNA in bone marrow samples using *in situ* PCR. Representative micrographs on detection of IS*6110* (corresponding to *Mtb* DNA) and IS*1311* (corresponding to *Mav* DNA) using *in situ* PCR in BMBs. (**a**) Macrophages (arrows) and sinusoidal endothelial cells (arrow heads) were the most common IS*6110* positive cells denoted by the cytoplasmic blue dots that contrasted with the nucleus stained by nuclear fast red. (**b**) Adipose cells also showed IS*6110* positivity (arrow), manifested by blue dots located in the cytoplasm pushed by large lipid vacuole. (**c**) Only occasional macrophages showed IS*1311* positivity (arrow), identified by cytoplasmic large dark-blue dot that contrasted with the red-stained nucleus. All micrographs are ×1000 magnification.

**Table 1 microorganisms-11-01788-t001:** Clinical data and results from group A biopsies.

	Clinical Records				IS*6110*		IS*1311*	
	TB History ^1^	HIV	Histology ^2^	Culture ^3^	SP + H	*is*-P	SP + H	*is*-P
BM1		+	NS	Mav	+	−	+	−
BM2		+	GM/AFB	Mtb	+	−	−	nd
BM4		+	NS	Mav	+	nd	+	nd
BM5		−	NS	Mtb	+	−	−	nd
BM7		+	NS	Mtb	+	−	−	−
BM8	history	+	GM	−	−	nd	−	nd
BM9	history	+	GM	Mav	+	−	+	−
BM10		+	NS	Mav	+	+	+	−
BM12	history	+	AFB	Mav	+	−	+	−
BM14	history	+	nd	Mav	+	−	−	−
BM16	history	+	GM	Mav	+	+	+	−
BM18	history	+	GM	−	+	nd	−	nd
BM19		+	NS	Mav	+	−	+	−
BM20	contact	+	GM	−	+	nd	−	nd
BM21		+	NS	Mtb	+	nd	+	nd
BM23 ^a^		+	NS	Mtb	+	+	−	−
BM24	history	+	GM	−	−	nd	−	nd
BM26	history	+	GM/AFB	Mav	+	−	+	−
BM27		+	NS	Mav	−	nd	+	nd
BM28		+	GM/AFB	Mtb	+	−	+	−
BM29 ^a^		+	GM	Mav	+	+	+	−
BM31 ^a^		+	NS	Mtb	+	+	+	−
BM33		+	MGM	Mav	+	−	−	−
BM34		+	NS	Mav	+	−	−	−
BM35		+	GM	Mav	+	+	+	−
BM36 ^b^		−	GM	Mtb	+	+	−	nd
BM37		+	GM	Mav	−	nd	−	nd
BM38		+	GM	Mtb	+	−	−	nd

^1^ Data relating to previous history of tuberculosis or recent contact with tuberculosis patients. ^2^ Histology: GM, granuloma; AFB, acid-fast bacilli; MGM, microgranuloma; NS, non-specific findings. ^3^ Culture: Mav, *M. avium*; Mtb, *M. tuberculosis*; −, negative. ^a^ Samples with *M. tuberculosis* DNA inside adipocytes using *in situ* PCR. ^b^ Patients on corticosteroid treatment at the time of sample collection. SP + H, standard PCR plus hybridization; *is*-P, *in situ* PCR. Results for PCR positive (+), negative (−), and not done (nd) are indicated.

**Table 2 microorganisms-11-01788-t002:** Clinical data and results from group B biopsies.

			IS*6110*		IS*1311*	
Sample	TB History ^1^	Histology ^2^	SP + H	*is*-P	SP + H	*is*-P
C1 ^a^		AML	+	+	+	nd
C2 ^a,b^		AML	−	+	+	+
C3		AML	−	−	−	nd
C4		AML	−	−	+	+
C5^a^		AML	−	+	+	+
C6		LS	−	nd	−	nd
C7		AML	−	−	+	−
C8		AML	+	+	−	+
C9	history	AML	+	+	−	−
C11		LS	−	−	+	−
C12	history	HA	−	−	+	−
C13		HA	+	+	−	+
C14	history	LS	−	−	−	−
C15	history	HA	−	−	−	nd
C16		HA	−	−	−	nd
C17 ^a^		HA	−	+	−	nd
C18		HA	−	−	−	nd
C19 ^b^		AML	−	−	+	−
C30		ST	−	nd	−	nd
C31		ST	−	nd	−	nd
C32		ST	+	+	−	+
C33	history	ST	+	+	−	−
C34		ST	+	+	−	+
C35		ST	+	+	−	−
C36		ST	−	−	−	−
C37		ST	−	−	−	−

^1^ Data related to previous history of tuberculosis. ^2^ Histology: AML, acute myeloid leukemia; LS, lymphoproliferative syndrome; HA, hematopoiesis alterations; ST, solid tumor. ^a^ Samples with *M. tuberculosis* DNA inside adipocytes using *in situ* PCR. ^b^ Patients on corticosteroid treatment at the time of sample collection. SP + H, standard PCR plus hybridization; *is*-P, *in situ* PCR. Results for PCR positive (+), negative (−), and not done (nd) are indicated.

## Data Availability

Not applicable.

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
