# Peer review of "Detection of Mycobacterial DNA in Human Bone Marrow"

_microorganisms, 2023, doi:10.3390/microorganisms11071788_

Round 1

Reviewer 1 Report

COMMENTS TO THE MANUSCRIPT Detection of Mycobacterial DNA in human bone marrow, by Alba González-Escalada et al.

General comment:

The submitted manuscript conducts a PCR + hybridization and in situ PCR in formalin-fixed paraffin-embedded human bone marrow samples to detect DNA from Mycobacterium tuberculosis and Mycobacterium avium species. Samples analyzed were form both HIV+ patients with mycobacterial infection and from subjects without evidence of infection. Results showed the DNA detection of mycobacterial species inside bone marrow macrophages. Based on this, the authors suggest that marrow adipocytes could be a reservoir for the persistence of M. tuberculosis, allowing the pathogen to establish long-lasting latent infection.

In general, the submitted manuscript is well written, the methodology is properly described, the results are clearly explained, and the discussion adequately addressed.

Thus, the document is suitable for its publication in the Microorganisms journal after a minor review.

Specific comments:

1. I suggest reallocating Figure 1 in the Results section.

2. Kindly indicate in the manuscript as Supplementary data the use of the probes for detecting the pathogens Klebsiella pneumoniae and Salmonella enterica if you will upload such archive to accompany the article.

3. The captions of the figures 1 and 2 must be improved to make all the figures self- explained. Add to the caption of the figure 1 a brief description of the controls and of each lane. Add to the caption of the figure 2 the meaning of arrows and arrowheads, the magnification of the microphotographs and the indication of a positive result. You describe all these in the main text near the figure, but as I previously stated, each figure must be self-explanatory. If you accept to do the modification of each caption, the main text of such results must be modified accordingly in order not to be redundant.

Please, review the use of commas in some parts of the text.

Author Response

Thanks very much for your revision and comments. They for sure are improving the quality of the new version of the manuscript.

Reviewer 2 Report

Alba González-Escalada et. al, have submitted a manuscript entitled ‘Detection of Mycobacterial DNA in human bone marrow’. The authors have performed the experiments very well and their result and conclusion are very clear from the manuscript. The study's results indicate a high prevalence of Mycobacterium tuberculosis (Mtb) infection in the group A biopsies, as evidenced by the detection of Mtb DNA in 24 out of 28 samples. Discrepancies between culture and PCR results suggest the presence of latent or non-culturable forms of Mtb. Mycobacterium avium (Mav) DNA was also detected, indicating Mav infection in the studied population. In situ PCR analysis revealed the cellular localization of Mtb and Mav DNA within various cell types. However, the differential distribution of bacilli and potential bone marrow involvement in asymptomatic infections may explain the observed discrepancies. "biopsies culture-positive for Mav" should be "biopsies positive for Mav in culture".Overall I have main concern with writing . there are  many errors ,but I have highlighted somes

1.    were biopsies with negative microbiological culture" should be "Two biopsies had negative microbiological culture".

2.    "histological analysis (these patients had a previous history of tuberculosis or a recent contact with tuberculous patient)" - The sentence structure seems incomplete. Consider revising it for clarity.

3.    "Some discrepant results were observed comparing culture and PCR results" should be "Some discrepant results were observed when comparing culture and PCR results"

4.    "non-culturable bacilli cannot be ruled out (e.g. BM18 or BM20)" - "BM18 or BM20" should be preceded by "such as" for clarity.

5.    "No mycobacterial DNA was detected in 3 of the samples in group A (BM8, BM24 192 and BM37)" - "BM8, BM24 192 and BM37" seems incorrect. Please verify and correct the sentence.

6.    "Positive detection of mycobacterial DNA from either Mtb or Mav (8/26 in each of them) was obtained in these samples by using PCR+H" should be "Positive detection of mycobacterial DNA, either Mtb or Mav (8/26 for each), was obtained in these samples using PCR+H."

7.    "The discrepancies detected when compared conventional versus in situ PCR using same target are a frequent finding in the literature and is usually explained by the differential distribution of the bacilli inside the tissues - "is" should be "are" to match the plural subject "discrepancies.Alba González-Escalada et. al, have submitted a manuscript entitled ‘Detection of Mycobacterial DNA in human bone marrow’. The authors have performed the experiments very well and their result and conclusion are very clear from the manuscript. The study's results indicate a high prevalence of Mycobacterium tuberculosis (Mtb) infection in the group A biopsies, as evidenced by the detection of Mtb DNA in 24 out of 28 samples. Discrepancies between culture and PCR results suggest the presence of latent or non-culturable forms of Mtb. Mycobacterium avium (Mav) DNA was also detected, indicating Mav infection in the studied population. In situ PCR analysis revealed the cellular localization of Mtb and Mav DNA within various cell types. However, the differential distribution of bacilli and potential bone marrow involvement in asymptomatic infections may explain the observed discrepancies. "biopsies culture-positive for Mav" should be "biopsies positive for Mav in culture".Overall I have main concern with writing . there are  many errors ,but I have highlighted somes

1.    were biopsies with negative microbiological culture" should be "Two biopsies had negative microbiological culture".

2.    "histological analysis (these patients had a previous history of tuberculosis or a recent contact with tuberculous patient)" - The sentence structure seems incomplete. Consider revising it for clarity.

3.    "Some discrepant results were observed comparing culture and PCR results" should be "Some discrepant results were observed when comparing culture and PCR results"

4.    "non-culturable bacilli cannot be ruled out (e.g. BM18 or BM20)" - "BM18 or BM20" should be preceded by "such as" for clarity.

5.    "No mycobacterial DNA was detected in 3 of the samples in group A (BM8, BM24 192 and BM37)" - "BM8, BM24 192 and BM37" seems incorrect. Please verify and correct the sentence.

6.    "Positive detection of mycobacterial DNA from either Mtb or Mav (8/26 in each of them) was obtained in these samples by using PCR+H" should be "Positive detection of mycobacterial DNA, either Mtb or Mav (8/26 for each), was obtained in these samples using PCR+H."

7.    "The discrepancies detected when compared conventional versus in situ PCR using same target are a frequent finding in the literature and is usually explained by the differential distribution of the bacilli inside the tissues - "is" should be "are" to match the plural subject "discrepancies.

Author Response

Thanks very much for your revision and comments.

Please see the attached file and the new version of the manuscript. We hope to accomplish all the requirements you mentioned that for sure are improving the quality of the manuscript.
